# Regional Variations in Urban Trash: Connections between Litter Communities and Place

**Randa L. Kachef and Michael A. Chadwick \***

Department of Geography, King's College London, 30 Aldwych, Bush House North East Wing, London WC2B 4BG, UK; randa.kachef@kcl.ac.uk

\* Correspondence: michael.chadwick@kcl.ac.uk

**Abstract:** Litter is a pervasive social and environmental issue that continues to evade effective and sustainable mitigation strategies. As the nature of waste items can influence methods and rates of littering, an understanding of litter typologies associated with specific sites has the potential to inform targeted anti-littering efforts. In this study, data analysis methods from ecology were applied to litter surveys to evaluate patterns among urban litter items found in two types of streets in England (High Streets and Central Business Districts). The results indicate that sites characterised as a High Street (predominantly leisure activities such as shopping and dining) contained lower densities and less variety yet featured litter items with a higher potential for environmental contamination than sites categorised as Central Business Districts (identified by high numbers of professional workers and transport links). Although litter was significantly different between sites, the litter community structure was not. Our results suggest that litter typologies and associated activities can lead to specific knowledge of key influential items in a site and inform future evidence-based and sustainable mitigation systems.

**Keywords:** litter; waste management; sustainability; community parameters; litter impact index

## 1. Introduction

Over half of the global population lives in an urban setting [1]. By congregating in a specific area, participants are benefited with symbiotic support between various working parts of a city. Like an ecosystem, urban spaces provide a singular point where complex security, education, employment, transport, and social networks are built. Designed to make life easier, the luxuries of living in an urban ecosystem are plenty. Although the importance of the urban area is universal, purpose and characteristics are unique to each—variations can be observed both between and within urban areas. These variations are often characterised by their primary use type, such as designations of business, leisure, and residential zoning.

Like any ecosystem, the urban ecosystem itself is vulnerable to stressors that threaten its wellbeing, one of these threats is the presence of litter. There are several social, financial and environmental issues associated with litter in urban spaces. The presence of litter is known to promote anti-social behaviour, leading to further littering [2], vandalism [3], and higher rates of crime and injury [4]. Littered communities typically experience a reduction in property values and tourist-derived income [4], while street cleansing costs divert funds from valuable statutory services such as libraries. Litter in the environment can attract and redistribute high levels of toxins and microbial pathogens [5,6], enable the long distance transportation of invasive species [7], and cause death to aquatic species through entanglement or ingestion [8,9]. Ultimately, 90% of marine plastics are suspected to originate from inland activities, transported to oceans via urban rivers [10]; addressing the issue of litter in urban centres has the potential to not only mitigate localised social, economic, and environmental repercussions but also stop the flow of plastics to marine environments.

Litter is a generalised term that encompasses a wide array of items. Despite distinguishable variations in typology, it is commonly accepted to be the byproduct of littering, a human behaviour [11]. Most often, the methods of and motivations for littering are specific to the waste items themselves [12,13], while specific activities produce different litter typologies. Equally, the material composition of litter items—for example, a straw made of plastic, which is easily transported in aquatic systems, versus one made of paper, which quickly begins to collapse when wet—determines the degree of associated negative social, economic, and environmental impacts [14].

Generally, litter mitigation efforts in urban spaces have focused on the individual, often seeking to influence littering behaviour [11,15]. This approach, however, does not consider individual spaces, variations between their use type, and the implications of these qualities on the associated litter typologies. Given that individual urban spaces are used for specific activities, it is theorised that the function of a street can influence the types of litter present. As littering behaviour is specific to the litter item itself, predicting litter typology profiles based on how a space is used can inform targeted approaches to mitigation; ultimately reducing the social, economic, and environmental impact associated with litter. As such, an understanding of site-specific litter profiles can inform targeted mitigation efforts.

Based on this understanding of litter and littering in urban spaces, and the lack of critical empirical evaluation of this, this study seeks to evaluate how different street types in different cities affect accumulation volumes and litter typology. Through systematic litter surveys, this study is designed to explore differences in litter profiles between regions and further grouping study sites by their purpose within the urban structure. Litter profiles will be compared in four ways: individual typology, prevalence, source activity, and impact potential.

## 2. Materials and Methods

### 2.1. Study Sites

To gain insight on regional variations in litter typology profiles, a series of systematic litter surveys were conducted in four sites located in the three largest cities in England. Specific sites within cities were chosen by local participating councils who identified their most problematic areas in terms of littering. The sites consisted of two London locations, Central: London Bridge (LBL) and South: Sutton High Street (SHS); Birmingham New Street (NSB); and Manchester Oxford Road (OXM) (Figure 1). The four sites fit into two distinct urban street types, the High Street (HS) and the Central Business District (CBD), and are all considered 'high intensity of use' in government land type indices [16].

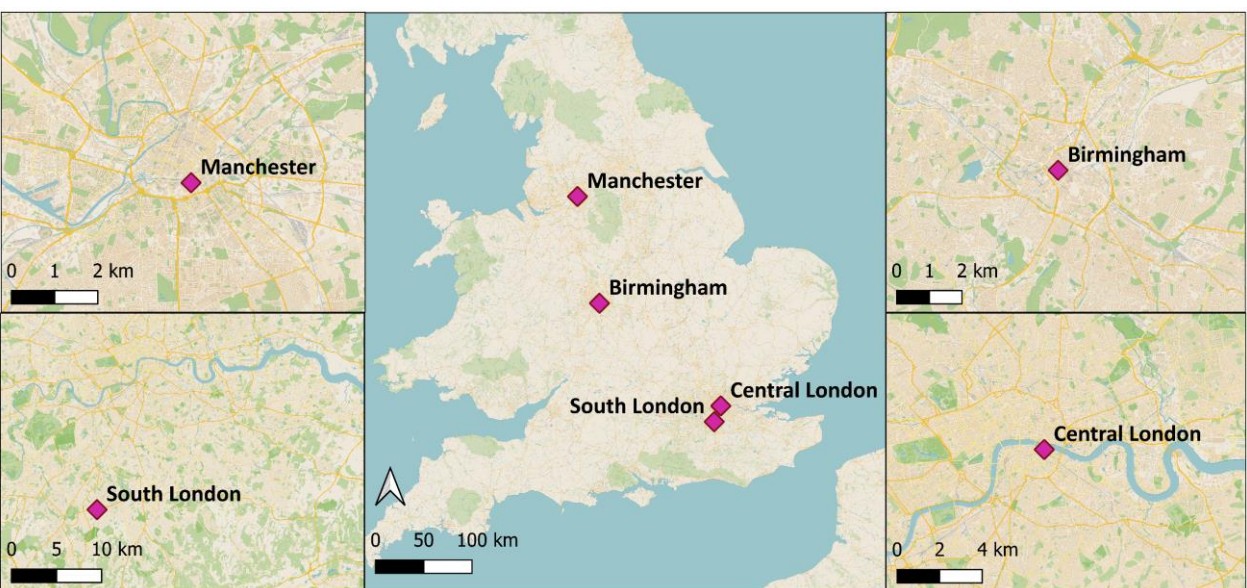

**Figure 1.** The four cities included in this study: Central London, South London, Birmingham, and Manchester.

Both the HSs (SHS and NSB) and the CBDs (OXM and LBL) are centrally located and highly connected in terms of transport, but the two serve very different roles in the lives of those who patronise them. The HS is a socio-economic hub, providing communal outdoor space for socialising, as well as a destination for food, shopping, and entertainment [17]. Whereas the distinguishing factor of a CBD is its role as a host to corporate office spaces and headquarters; often attracting healthcare and higher education institutions, it is patronised by a large population of highly skilled workers [18,19] and exhibits a peak use time in accordance with weekday working hours.

### 2.2. Litter Survey

Prior to data collection sessions, sample sites were canvassed and an inventory of present litter was taken. This inventory was used to inform templated data collection forms and ensure that as many litter typologies as possible were included, whilst being cognisant that, for ease of reporting, the form should fit on a single page. This was carried out with the intention of reducing the portion of data in the sample classified as 'other'.

During data collection sessions, each site was divided in sectors, allowing for multiple individuals to act as counters, simultaneously collecting data while eliminating any potential for duplicate counts. Sites varied in size and the number of sectors depended on number of counters available to conduct the survey simultaneously (Figure 2).

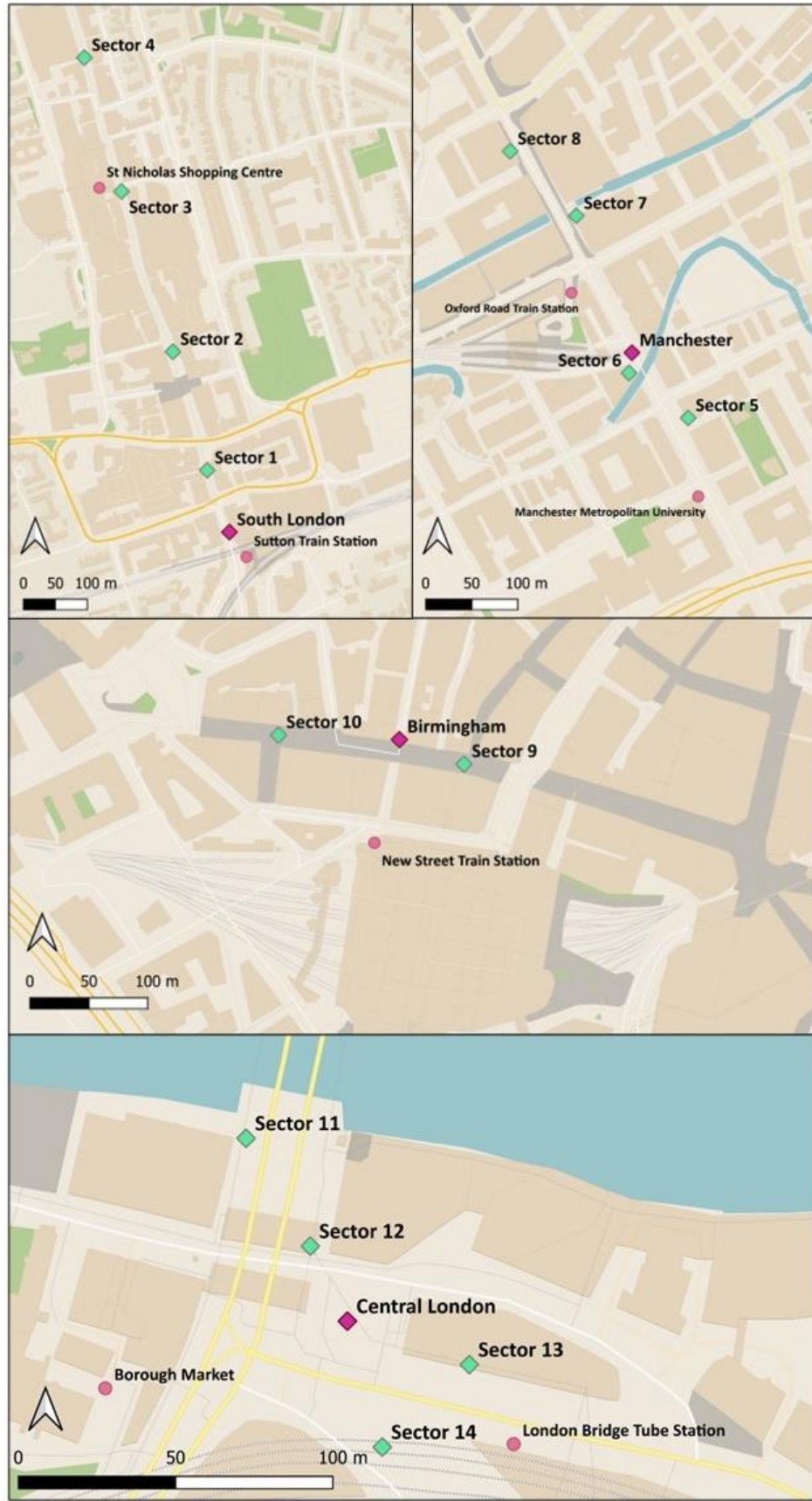

**Figure 2.** Location of sectors within each study site.

Additional requirements for sectors were that they could each be surveyed in their entirety within a similar and reasonable timeframe, approximately 30 min. Due to the existing nature of the study sites, each varied in total area and distance (Table 1).

**Table 1.** Details on size, length, and sector divisions of study sites.

| City | Total Area (sq/m) | Number of Sectors | Mean Sector Area (sq/m) | Length of Study Site (m) |
|---|---|---|---|---|
| LBL | 3034 | 4 | 758 | 321 |
| SHS | 10,717 | 4 | 2679 | 556 |
| NSB | 5155 | 2 | 2578 | 350 |
| OXM | 3458 | 4 | 864 | 508 |

To ensure consistency across all counters, prior to data collection sessions, each counter was guided through the extent of their sector and trained in litter identification. During these exercises, data collection forms were reviewed to guarantee counters were familiar with and understood the meaning of each litter typology. Observers were often local to each study site (i.e., observers in London did not collect data in Manchester, etc.); therefore, emphasis on comprehension of and consistency in data collection methods was paramount.

Counters were provided with data collection forms (Figure A1 in Appendix A) that included 32 different litter typologies (Figure 3), allowing for a high-resolution tally method of data. Typologies were both informed by initial site visits and borrowed from previous studies, e.g., Keep Britain Tidy, 2020 [8], allowing for consistency within the field of research and comparative analysis if desired. Additionally, counters were provided with detailed maps of their designated counting areas, which included notes on physical markers such as lamp posts and changes in pavement tiles to aid in identifying area boundaries. As the health and safety of counters was of the upmost concern, instructions were to never count littered items located on a roadway and to only count litter items present in pedestrian designated areas.

Depending on the number of sectors in the study site, 2 to 4 counters collected data simultaneously. During data collection sessions, counters were instructed to systematically canvas the entirety of their sector, tallying each item of litter they encountered on the data collection form. This was achieved by walking in a sweeping pattern across the width of the street from one end of their designated area to the other. Each sector took no more than 30 min to canvass in its entirety. To remain within research budgets, each site was counted on 4 separate occasions.

| Activity | Cigarettes | Smoking | Gum | Drinking | Eating | Transport | Shopping and entertainment | Other |
|---|---|---|---|---|---|---|---|---|
| Typology | Cigarette end | Cigarette packaging, cellophane wrap, foil, rolling paper, unsmoked filter, lighter and match. | Chewing gum in 3D form. | Plastic bottle, tin can, paper cup, hot drink cup, insulating wrap, cup lid, glass bottle, straw and other drink items. | Crisp or chip bag, sweet wrap or bag, takeaway box, polystyrene box or tray, sandwich pack or wrap, napkin, tissue, paper bag, utensil, cellophane wrap, food and other food related items. | Ticket: bus, train, plane, tube, tram, etc. | Cash point and ATM receipt, till receipt, flyer, leaflet, carboard box, newspaper, magazine and plastic bag. | Textile, bagged litter, general, unsure and other. |

**Figure 3.** Definition of litter typologies grouped by source activity.

*2.3. Analysis*

To ascertain if the individual spaces have unique litter profiles, data analysis techniques drawn from community ecology were applied to the surveyed litter items and their Litter Impact Index values (LIIV) [14]. Community relationships (see specific approaches below) were generally considered indicative measurements of the strength and diversity of a "litter" ecosystem [20] and were used here in the context of understanding the structure and diversity of specific litter typologies within the study sites [21]. Additional parameters not only include measurements of volume, a traditional approach to litter analysis, but analysed variety in items present and identifies dominant typologies within each site. For the purposes of this study, each litter item was considered as an analogue to species in a community. Additionally, litter items with closely related purposes were further grouped at a higher typological level (e.g., like Genera in Linnaean taxonomic classification). In the context of this study, this was defined by the source activity that created them (e.g., drinking: bottles, cups, straws, etc.); a total of 8 source activities were established (Figure 3). Cigarettes were designated their own source activity and not grouped within the smoking genus as their abundance led to inflated values of other smoking-related items such as packaging, rolling papers, and unsmoked filters. Equally, as chewing gum is consumed under a variety of scenarios (e.g., after smoking and eating, as a snack, out of habit, to freshen breath, etc.), the source activity could not be assumed and was designated its own source activity. Together this approach allowed us to create a specific litter typology (Figure 3).

Litter items were recorded by typology and source activity within each sector from 4 separate counting sessions. Sectors by counting session were considered independent sampling units and, except for areas designated as roadways, were surveyed in their entirety.

A series of indices on typology were calculated for each sampling unit and are considered community parameters. Community parameters (CPs) include abundance (total number of litter items found), richness (number of litter typologies present), density (abundance divided by subsector area), evenness (richness divided by abundance), and impact (mean LIIV of counted items). Due to the nature of the sampling units, the summary counts of the data were assumed non-parametric and were not transformed for normality.

As counting sessions were not simultaneous across all sites, the influence of different sessions was first calculated to establish if time and date were factors in CP similarity analysis. This was analysed in SigmaPlot (version 14.5), using Kruskal–Wallis one-way ANOVA on ranks comparing means with the Tukey Test [22,23]. The influence of time and date has been considered inconsequential in similar studies [24].

As the size of sectors were not consistent, abundance was not included in community parameter analysis, and analysis of density was considered as an appropriate indicator of volumes of litter present. Regional influences on community parameters (richness, density, evenness, and impact) were analysed using a Kruskal–Wallis Test between cities (SHS, OXM, NSB, and LBL). When significant differences were observed, a Dunn's method pairwise comparison procedure was employed to further investigate. The influence of street types (HS and CBD) on community parameters was analysed using a Wilcoxon Test. In all comparison analyses, differences were considered statistically significant when $p < 0.05$. Violin plots to represent community parameters by city and street type were built in R Studio (version 2023.03.2) using the ggplot2 (version 3.3.6) package.

The overall diversity of typologies by city and street type were calculated in Excel (version 16.0) using the Shannon Diversity Index then transformed for ease of comprehension to the Shannon Equitability Index (Q) [25]. Data on litter typology and source activity were normalised by area and modelled in R Studio (version 2023.03.2) using the vegan (version 2.6-2) and cluster (version 2.1.3) software packages. An Analysis of Similarity (ANOSIM) was run to test for similarities in litter composition among city- and street-type groupings [26,27] and the Similarity Percentage analysis (SIMPER) procedure was employed to identify particular typologies and activity groupings that were responsible for these similarities [26,28]. Finally, a non-metric multidimensional scaling (nMDS) model

was performed using Bray–Curtis distances to illustrate community structures within test sites [29].

## 3. Results

During four separate data collection sessions in each of the four sites, a total of 26,209 items of litter were counted within 132 subsectors. Data were collected on the following dates: SHS: 4, 5, 11, and 12 March 2016; OXM: 19, 20, 25, and 27 May 2016; NSB: 3, 4, 10, and 11 June 2016; LBL: 6, 8, 20, and 22 April 2017. Litter items were categorised into 32 typologies and 8 source activities (Table A1 in Appendix B). Note that due to construction, a small portion of OXM (sector 7, subsector 45) on 25 May 2016 (session 3) was inaccessible and, therefore, not counted.

### 3.1. Influence of Sessions

The results identify differences in richness ($p \leq 0.001$), yet none by density ($p = 0.051$), evenness ($p = 0.489$), or impact ($p = 0.515$). Given the similarity in richness medians and evidence from prior research, it was decided to discount the influence of sessions and establish that time and date had no significant impact on litter survey results.

### 3.2. Community Parameters

Results of CP both by city and pooled by street type are seen in Figure 4. The overall shape of a violin represents the distribution of data while individual dots represent actual data points. Box and whisker plots within the violin represent interquartile intervals. With the exception of the richness parameter, general violin shapes indicate higher levels of distribution among CBD sites (OXM/LBL) with a concentrated pattern among HS sites (SHS/NSB).

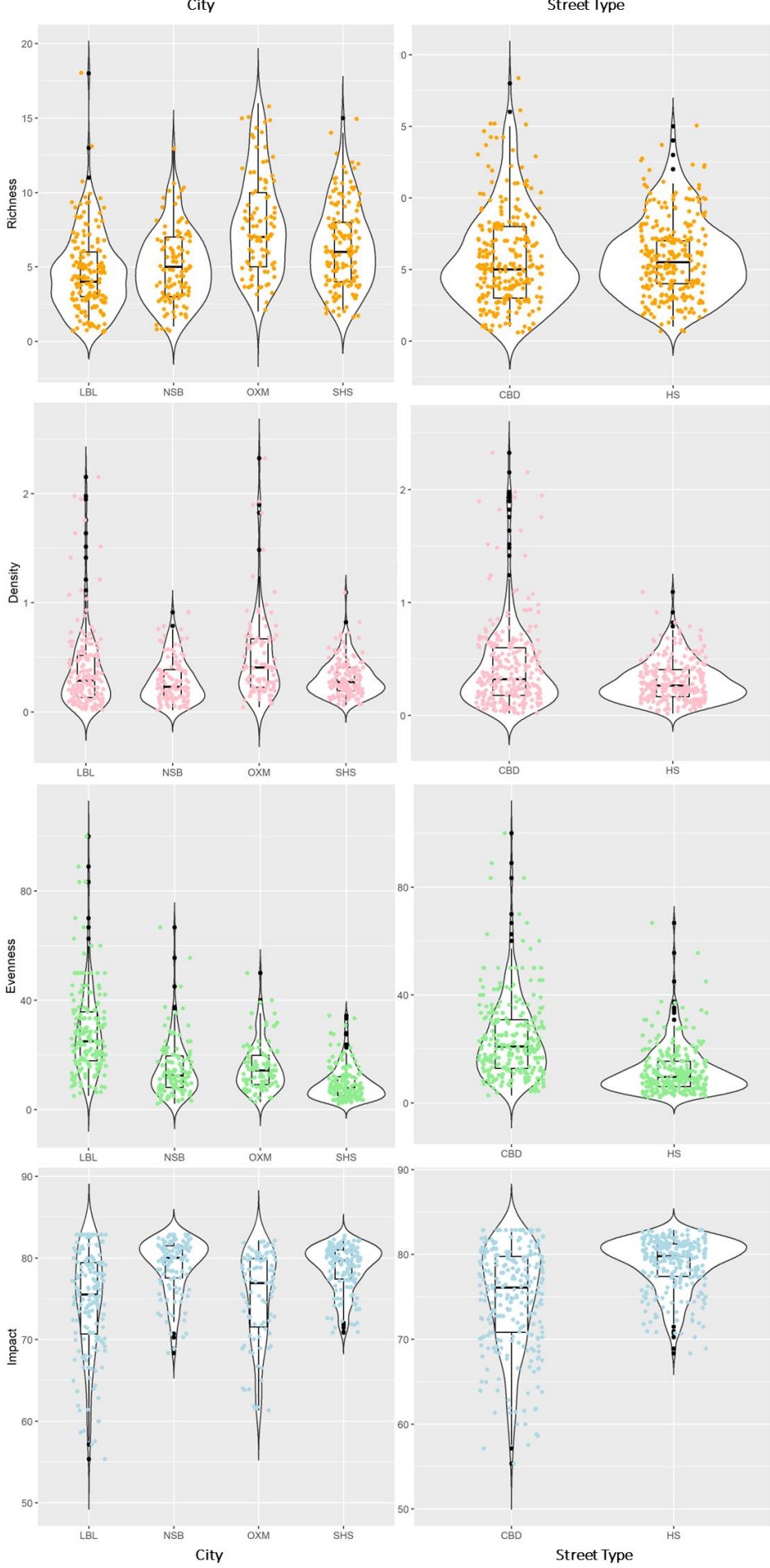

**Figure 4.** Community parameter distribution pooled by city and street type.

### 3.3. Regional Influence

A one-way ANOVA between community parameters and sites tested for regional differences (Table 2). It was found that all parameters were significantly different between cities. When pooled by street type, however, there were similarities within the richness parameter, which is consistent with patterns observed in violin plots in Figure 4.

**Table 2.** One-way ANOVA of community parameters between cities (Kruskal–Wallis) and street type (Wilcoxon).

| Parameter | City | Street Type |
|---|---|---|
| Richness | $p < 0.001$ | $p = 0.427$ |
| Density | $p < 0.001$ | $p = 0.003$ |
| Evenness | $p < 0.001$ | $p < 0.001$ |
| Impact | $p < 0.001$ | $p < 0.001$ |

An all pairwise mean comparison analysis was run between parameters within city pairings (Table 3). Strong city groupings were found within the density parameter between both London locations (SHS/LBL); within the evenness parameter between both locations outside of London (OXM/NSB); and within the impact parameter between HS locations (SHS/NSB) and CBD locations (OXM/LBL).

**Table 3.** Dunn's method of all pairwise mean comparison of parameters between cities.

| Parameter | | SHS | OXM | NSB |
|---|---|---|---|---|
| | OXM | $p = 0.023$ | - | - |
| Richness | NSB | $p = 0.002$ | $p < 0.001$ | - |
| | LBL | $p < 0.001$ | $p < 0.001$ | $p = 0.915$ |
| | OXM | $p < 0.001$ | - | - |
| Density | NSB | $p = 0.712$ | $p < 0.001$ | - |
| | LBL | $p = 1$ | $p < 0.001$ | $p = 0.617$ |
| | OXM | $p < 0.001$ | - | - |
| Evenness | NSB | $p < 0.001$ | $p = 1$ | - |
| | LBL | $p < 0.001$ | $p < 0.001$ | $p < 0.001$ |
| | OXM | $p < 0.001$ | - | - |
| Impact | NSB | $p = 1$ | $p < 0.001$ | - |
| | LBL | $p < 0.001$ | $p = 1$ | $p < 0.001$ |

### 3.4. Diversity

To establish diversity of litter typologies, the Shannon's Equitability Index (Q) values were calculated (Figure 5). The diversity within cities clearly shows higher diversity in CBDs (OXM/LBL) while HSs (SHS/NSB) feature lower diversity scores.

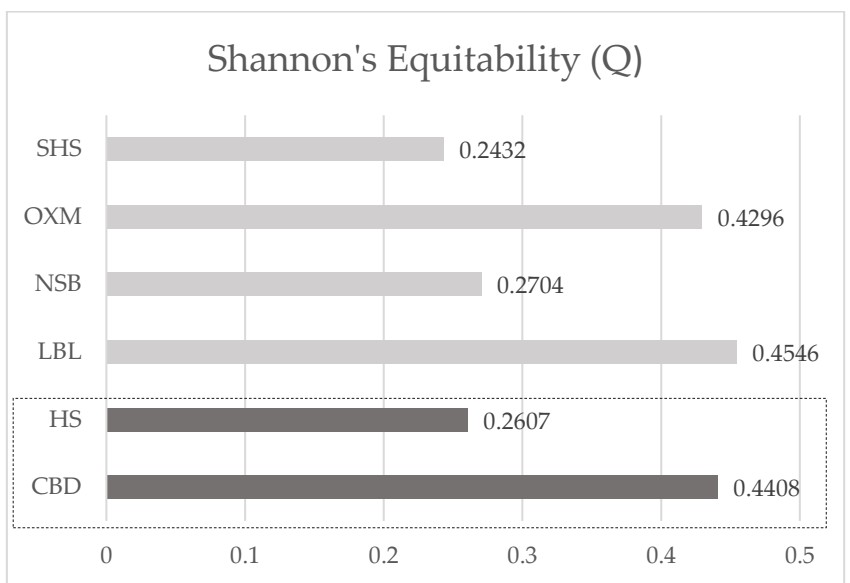

**Figure 5.** Shannon's Index of Equitability values depicting diversity of litter typology by city (light grey) and street type (dark grey). The results range from 0 to 1, where lower values represent communities with the fewer species.

### 3.5. Litter Composition

The Analysis of Similarity (ANOSIM) results in an R value that is measured on a scale of 0–1, where a high value represents greater differences in litter typology composition between sites [26]. Values were exceptionally low in comparisons of litter typologies between cities (R = 0.04263, $p < 0.001$) and street types (R = 0.03735, $p < 0.001$), implying high similarities between typology composition. Equally, low values were observed in comparisons of litter activity groupings between cities (R = 0.04172, $p < 0.001$) and street types (R = 0.04449, $p < 0.001$), implying high similarities between activity grouping composition.

The Similarity Percentage analysis (SIMPER) output in Table 4 represents similarity in litter typology and activity groups by city pairings and street types. In terms of typologies, similarity values were highest between CBD sites LBL and OXM (58.57%), while the lowest similarities were found between the HS sites SHS and NSB (46.02%). The overall percentage of litter typology similarity explained by street types (HS/CBD) was 53.05%. When grouped by activity, similarity values were again highest between CBD sites LBL and OXM (53.93%) but also between LBL and NSB (52.38%), while the lowest similarities observed were between HS sites SHS and NSB (42.89%). The overall percentage of similarity of activity groupings explained by street types (HS/CBD) was 49.59%.

**Table 4.** SIMPER percentage of litter typology and activity grouping similarity between cities and street types.

| Parameter | City | SHS | OXM | NSB | HS |
|---|---|---|---|---|---|
| Typology | OXM | 49.38% | - | - | - |
| | NSB | 46.02% | 54.18% | - | - |
| | LBL | 52.54% | 58.57% | 55.86% | - |
| | CBD | - | - | - | 53.05% |
| Activity | OXM | 45.71% | - | - | - |
| | NSB | 42.89% | 50.25% | - | - |
| | LBL | 49.40% | 53.93% | 52.38% | - |
| | CBD | - | - | - | 49.59% |

To better understand which litter typologies and activity groupings contributed towards similarity values between street types, Table 5 lists typologies in order of greatest influence. Unsurprisingly cigarette ends were the most influential litter typology (58%),



followed by chewing gum (6%). Cigarettes, again, were the most influential activity grouping (63%), followed by eating (10%).

**Table 5.** Influence of litter typology and activity groupings towards similarity between street types HS and CBD (SIMPER).

| Parameter | Item | Contribution |
|---|---|---|
| Typology | Cigarette end | 58% |
| | Chewing gum in 3D form | 6% |
| | General litter & other | 3% |
| | Sweet wrap or bag | 3% |
| | Tissue & napkin | 3% |
| | Flyer and leaflet | 3% |
| | Till receipt | 2% |
| | Cigarette packaging | 2% |
| | Cash point & ATM receipt | 2% |
| | Food | 2% |
| | Train or bus ticket | 2% |
| | Plastic bottle | 1% |
| | Cellophane wrap | 1% |
| | Takeaway box | 1% |
| | Utensil | 1% |
| | Tin or can | 1% |
| | Cigarette rolling paper & unsmoked filter | 1% |
| | Unsure | 1% |
| | Sandwich pack and wrap | 1% |
| | Paper cup for cold drink | 1% |
| | Other drink item | 1% |
| | Newspaper and magazine | 1% |
| | Paper bag | 1% |
| | Cardboard box | 1% |
| | Lighter and match | 1% |
| | Hot drink cup | 0% |
| | Crisp or chip bag | 0% |
| | Glass bottle | 0% |
| | Polystyrene food box or tray | 0% |
| | Bagged litter | 0% |
| | Plastic bag | 0% |
| | Textile | 0% |
| Activity | Cigarette | 63% |
| | Eating | 10% |
| | Shopping & Entertainment | 7% |
| | Gum | 6% |
| | Other | 5% |
| | Drinking | 4% |
| | Smoking | 4% |
| | Transport | 1% |

Activity groups were analysed between city pairs to identify those with the greatest influence (Table 6). Similar to street-type comparisons, cigarette ends were most influential across all pairings followed by eating.

**Table 6.** Activity grouping contribution towards similarity between pairs of cities (SIMPER).

| LBL/OXM Overall Similarity: 53.93% | | LBL/SHS Overall Similarity: 49.40% | |
|---|---|---|---|
| Cigarette | 58% | Cigarette | 63% |
| Eating | 11% | Eating | 11% |
| Shopping & Entertainment | 9% | Gum | 6% |
| Other | 5% | Shopping & Entertainment | 7% |
| Gum | 6% | Other | 5% |
| Drinking | 5% | Drinking | 4% |
| Smoking | 4% | Smoking | 4% |
| Transport | 2% | Transport | 0% |
| **LBL/NSB overall similarity: 52.38%** | | **OXM/SHS overall similarity: 45.71%** | |
| Cigarette | 62% | Cigarette | 62% |
| Eating | 12% | Eating | 10% |
| Shopping & Entertainment | 7% | Shopping & Entertainment | 8% |
| Gum | 6% | Other | 7% |
| Drinking | 5% | Drinking | 4% |
| Smoking | 4% | Gum | 3% |
| Other | 3% | Smoking | 3% |
| Transport | 1% | Transport | 3% |
| **OXM/NSB overall similarity: 50.25%** | | **NSB/SHS overall similarity: 42.89%** | |
| Cigarette | 62% | Cigarette | 72% |
| Eating | 10% | Eating | 7% |
| Shopping & Entertainment | 9% | Shopping & Entertainment | 7% |
| Other | 5% | Other | 4% |
| Drinking | 5% | Gum | 4% |
| Smoking | 3% | Smoking | 3% |
| Gum | 3% | Drinking | 3% |
| Transport | 3% | Transport | 0% |

*3.6. Community Structure*

The non-metric multidimensional scaling model (nMDS) is interpreted in stress values, where those >0.3 suggest clustered patterns and <0.2 imply a weak random relationship [25]. The analysis on litter typology resulted in a stress value of 0.11, and the data are, therefore, considered random. Figure 6 illustrates the output, where HSs are mostly clustered within the centre, while CBDs sites exhibit a wider spread.

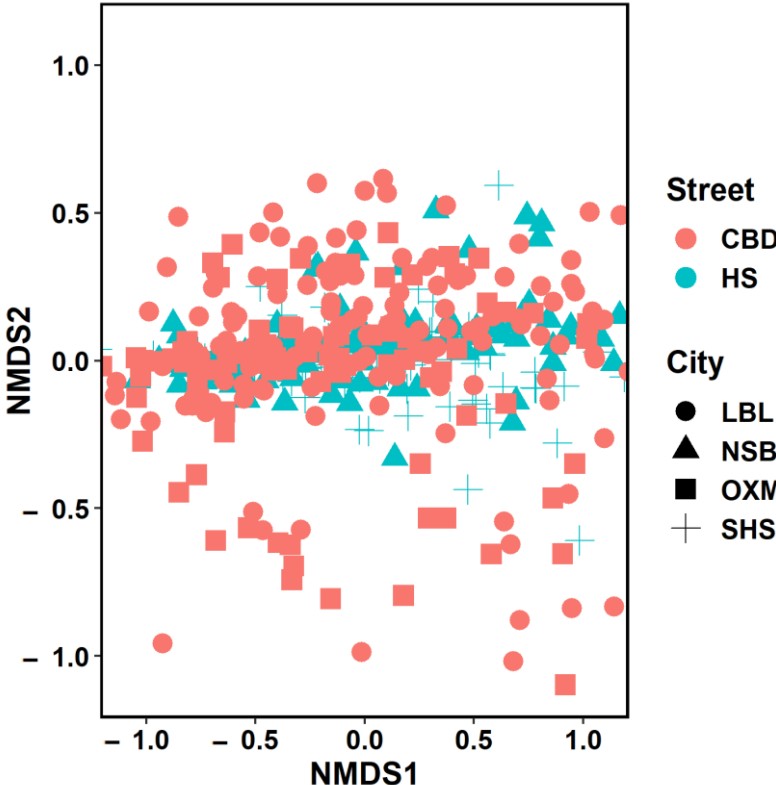

**Figure 6.** NMDS on litter typology is considered random.

The nMDS on activity groupings resulted in a stress value of 0.1, and the data are, therefore, considered random. Figure 7 illustrates the output and no clearly defined community clusters are apparent, with sites exhibiting wider spreads from the centre.

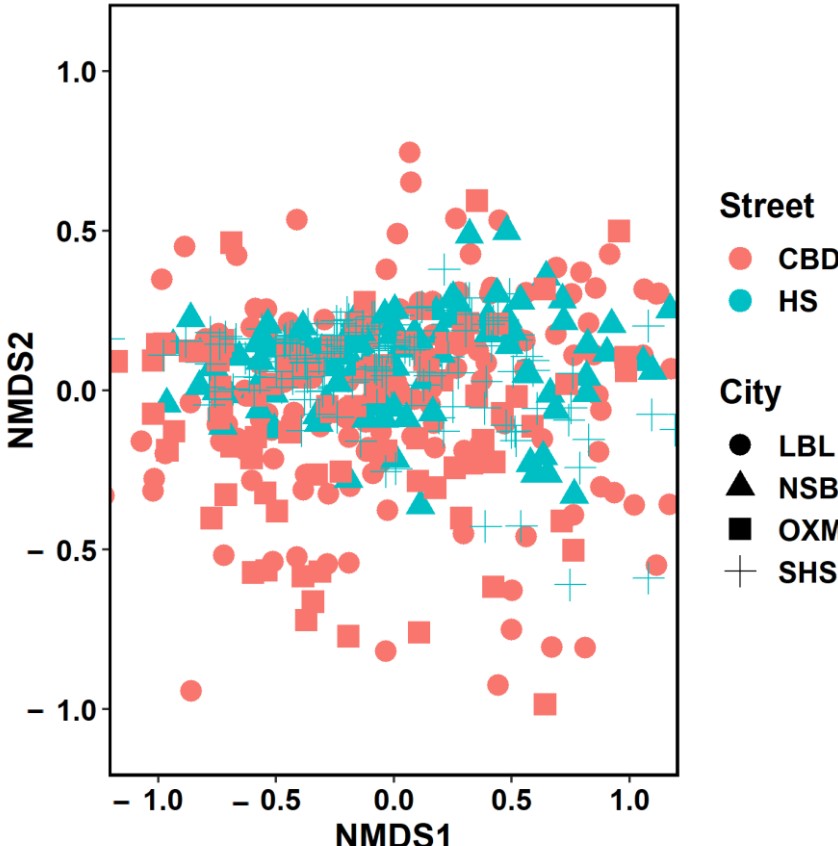

**Figure 7.** NMDS on litter activity groupings is considered random.

### 3.7. Summary of Results

This study examined patterns in litter typology and source activity between four English cities (SHS, OXM, NSB, and LBL), as well as within two street type groupings, the High Street (HS: SHS and NSB) and the Central Business District (CBD: OXM and LBL). Data were collected in each site during four sessions, and in line with previous research, the time and date of sessions were considered to have no influence on litter patterns [30].

Violin plots on density and impact indicate strong similarities in data distribution between street-type groupings, while richness and evenness violin plots suggest the same, yet to a lesser degree. Generally, the results suggest community parameters (CPs) were influenced by street type, with CBD sites exhibiting higher densities of varied items (Q = 0.44) with lower impact values (as seen in Figure 4) and HS sites featuring fewer items (Q = 0.26) of higher impact values (as seen in Figure 4), with specific typologies dominating the sample (e.g., cigarettes accounting for 58% of similarity in typology profile).

Significant differences ($p < 0.001$) were found by city within all CPs. In pairwise mean comparisons, similarities were found in richness between NSB/LBL; in density between NSB/LBL, SHS/NSB, and LBL/SHS; in evenness between OXM/NSB; and in impact between SHS/NSB and OXM/LBL. When pooled by street type, no difference was found in richness ($p = 0.427$), yet significant differences were found in density ($p = 0.003$), evenness ($p < 0.001$), and impact ($p < 0.001$). The results suggest that there are some connections between street type and CP regarding volumes of litter and the level of impact associated with typologies present.

Diversity is highly influenced by factors of richness and evenness and the results of the Shannon's Equitability Index were in line with observations on community parameters. Both the HS (0.26) sites, SHS (0.24) and NSB (0.27), scored low on diversity, while the CBD (0.44) sites, OXM (0.45) and LBL (0.43), were considerably higher. The results suggest a more varied and equal distribution of litter typologies in CBDs.

Generally, litter composition was similar across sites in both typology and activity groupings. Highest similarities were found between LBL/OXM (CBD sites) in both litter typology (58.57%) and activity groupings (53.93%), while the lowest similarities were found between SHS/NSB (HS sites) in both categories (typology 46.02%; activity 42.89%). In typology the largest contributor to similarity was, unsurprisingly, cigarette ends (58%), followed by chewing gum (6%), while in activity groupings, cigarette ends and eating were the highest contributors to similarity. When grouped by street type, similarities in typology (53.05%) were higher than that observed in activity groupings (49.59%). The results suggest that cigarette ends were the most influential species, and although CBD sites had similarities in composition, HS sites did not. The data suggest that cigarette ends were more frequently observed in HS sites where leisurely activities were taking place. Cigarette ends are an item that are not only littered more frequently but are disposed of in uniquely specific ways [21,22].

No distinguishable typology or activity groupings community structure were found, although the nMDS indicated spread was wider among CBD sites than HSs.

## 4. Discussion

As is seen all over the world, urbans areas have a set of unique and distinguishable characteristics. This applies not only to the language and aesthetics of places but can also apply to cultural and individual values, particularly in relation to consumption and post-consumption behaviour. Societal, economic, and environmental problems associated with litter are plenty; however, quantitative studies to evaluate numbers in urban spaces are limited.

This study found that the density, diversity, and impact of litter items are influenced by the purpose of a street and, to a lesser degree, the items that are present. Sites that were designated CBDs typically contained a wider range of items, whereas HSs were characterised by fewer dominant typologies. Although litter typologies and composition within CBD sites were similar, those in HSs do not appear to be predictable.

Aspects of uncertainty are present in the use of evaluation parameters such as richness, evenness, and diversity, where specifics of community structure can be overlooked through the quantification of individual traits [20]. This study avoids this pitfall through the inclusion of Litter Impact Index values, lending weight to parameters and signalling differences in regional communities. Through the impact weighting of litter typologies, opportunities arise to establish mitigation systems that specifically target places with highly toxic and environmentally detrimental litter profiles. For example, higher density rates in CBDs would suggest that CBDs be the focus for environmental enhancement initiatives. However, despite CBDs containing more litter items by density, impact values were significantly lower than those found in HSs. As a result, the use of targeted initiatives in HSs against high-impact litter items can reduce associated environmental impact, while initiatives to mitigate the social and economic impacts of litter would be more effective by targeting the higher volumes of litter associated with CBDs.

This comparison of litter profiles between areas of high intensity of use can inform decision makers on local and national litter trends by identifying specific items and activities of concern. Previous research has found that the use of direct cues and tailored approaches focused on specific litter types has considerably higher success than generalised messages of 'do not litter' [23–25]. By establishing litter profiles specific to a site and how it is used, targeted mitigation initiatives can be developed. For example, an 11% similarity in typologies associated with eating was observed in the CBD. This suggests that efforts by local authorities could be more efficient by adjusting cleansing strategies in those areas to cater to meal timings or by establishing a localised deposit return scheme for take away packaging. Equally, in HSs, not only were impact scores significantly higher but a 7% similarity among litter items associated with shopping was also observed. In this scenario, national-level interventions to reduce the use and distribution of thermal papers (until and cash point receipts) could be implemented. Across all sites, however, cigarette end

litter was responsible for the highest levels of similarities and impact, indicating a broader and more urgent need to address their use and associated legislation.

By framing litter as a product of place and not one of human behaviour, this study argues that sustainable waste management systems should be informed on a local level, adjusting to the specific typology profiles of individual areas. Given the current concerns associated with litter, a reduction in volumes by specifically targeting dominant items is a means to alleviate social, economic, and environmental impacts, as well as their contribution to global issues of marine plastics.

**Author Contributions:** Conceptualization, R.L.K.; methodology, M.A.C. and R.L.K.; software, M.A.C. and R.L.K.; validation, R.L.K. and M.A.C.; formal analysis, R.L.K. and M.A.C.; investigation, R.L.K.; resources, R.L.K.; data curation, M.A.C. and R.L.K.; writing—original draft preparation, R.L.K.; writing—review and editing, M.A.C. and R.L.K.; visualisation, R.L.K. and M.A.C.; supervision, M.A.C.; project administration, R.L.K.; funding acquisition, R.L.K. All authors have read and agreed to the published version of the manuscript.

**Funding:** This research was funded in part by the Hubbub Foundation UK, registered charity no. 1158700.

**Institutional Review Board Statement:** Not applicable.

**Informed Consent Statement:** Not applicable.

**Data Availability Statement:** Litter typology data and mapping shapefiles for this study are freely available at https://doi.org/10.18742/19463102 (accessed on 30 May 2024).

**Acknowledgments:** Many thanks to David Demeritt who was the inspiration in envisioning a street and the litter present as a river and its ecology.

**Conflicts of Interest:** The authors declare no conflicts of interest.

## Appendix A

| | Name, date, time, sector and weather | | | | | |
|---|---|---|---|---|---|---|
| | Subsector | | | | | |
| | *Litter type* | *Count* | | | | |
| Cigarette material | Cigarette end | | | | | |
| | Cigarette packaging, cellophane wrap and foil paper | | | | | |
| | Cigarette paper and unsmoked filter | | | | | |
| | Lighter and match | | | | | |
| | Chewing gum in 3D form | | | | | |
| Drinks | Plastic bottle | | | | | |
| | Tin or can container | | | | | |
| | Hot drink cup | | | | | |
| | Hot drink insulating wrap | | | | | |
| | Paper cup for cold drink | | | | | |
| | Glass bottle | | | | | |
| | Other drink item | | | | | |
| Food packaging | Crisp or chip packet | | | | | |
| | Sweet wrap or bag | | | | | |
| | Takeaway box made of card, plastic and aluminium | | | | | |
| | Polystyrene food box | | | | | |
| | Sandwich pack and wrap | | | | | |
| | Tissue and napkin | | | | | |
| | Paper bag | | | | | |
| | Utensil | | | | | |
| | Food | | | | | |
| | Cellophane wrap | | | | | |
| | Other food item | | | | | |
| Shopping and Entertainment | Train or bus ticket | | | | | |
| | Cash point or ATM receipt | | | | | |
| | Till receipt | | | | | |
| | Flyer and leaflet | | | | | |
| | Cardboard box | | | | | |
| | Newspaper and magazine | | | | | |
| | Plastic bag | | | | | |
| Other | Textile | | | | | |
| | General litter and other | | | | | |
| | Unsure | | | | | |
| | Bagged litter | | | | | |
| | TOTAL COUNT OF LITTER: | | | | | |

**Figure A1.** Litter typology data collection form.

## Appendix B

**Table A1.** Results of data collection sessions, including litter typologies and activity groupings.

| Item | Activity | Total | % | SHS | % | OXM | % | NSB | % | LBL | % |
|---|---|---|---|---|---|---|---|---|---|---|---|
| Cigarette end | Cigarette | 20,640 | 78.75% | 9919 | 84.20% | 4279 | 70.44% | 4063 | 82.85% | 2379 | 68.96% |
| Cigarette packaging, cellophane wrap and foil paper | Smoking | 323 | 1.23% | 53 | 0.45% | 88 | 1.45% | 115 | 2.35% | 67 | 1.94% |
| Cigarette rolling paper and unsmoked filter | Smoking | 103 | 0.39% | 30 | 0.25% | 34 | 0.56% | 0 | 0.00% | 39 | 1.13% |
| Lighter and match | Smoking | 77 | 0.29% | 34 | 0.29% | 15 | 0.25% | 2 | 0.04% | 26 | 0.75% |
| Chewing gum in 3D form | Gum | 561 | 2.14% | 215 | 1.83% | 129 | 2.12% | 52 | 1.06% | 165 | 4.78% |
| Plastic bottle | Drinking | 149 | 0.57% | 36 | 0.31% | 52 | 0.86% | 25 | 0.51% | 36 | 1.04% |
| Tin or can container | Drinking | 106 | 0.40% | 21 | 0.18% | 42 | 0.69% | 12 | 0.24% | 31 | 0.90% |
| Paper cup for cold drink | Drinking | 92 | 0.35% | 40 | 0.34% | 25 | 0.41% | 20 | 0.41% | 7 | 0.20% |
| Hot drink cup | Drinking | 86 | 0.33% | 21 | 0.18% | 31 | 0.51% | 4 | 0.08% | 30 | 0.87% |
| Glass bottle | Drinking | 54 | 0.21% | 7 | 0.06% | 31 | 0.51% | 4 | 0.08% | 12 | 0.35% |
| Other drink item | Drinking | 58 | 0.22% | 7 | 0.06% | 3 | 0.05% | 33 | 0.67% | 15 | 0.43% |
| Crisp or chip packet | Eating | 85 | 0.32% | 17 | 0.14% | 45 | 0.74% | 8 | 0.16% | 15 | 0.43% |
| Sweet wrap or bag | Eating | 470 | 1.79% | 200 | 1.70% | 108 | 1.78% | 67 | 1.37% | 95 | 2.75% |
| Takeaway box made of card, plastic and aluminium | Eating | 160 | 0.61% | 43 | 0.37% | 86 | 1.42% | 19 | 0.39% | 12 | 0.35% |
| Polystyrene food box or tray | Eating | 42 | 0.16% | 10 | 0.08% | 12 | 0.20% | 5 | 0.10% | 15 | 0.43% |
| Sandwich pack or wrap | Eating | 93 | 0.35% | 11 | 0.09% | 40 | 0.66% | 23 | 0.47% | 19 | 0.55% |
| Tissue or napkin | Eating | 376 | 1.43% | 159 | 1.35% | 80 | 1.32% | 54 | 1.10% | 83 | 2.41% |
| Paper bag | Eating | 44 | 0.17% | 10 | 0.08% | 17 | 0.28% | 3 | 0.06% | 14 | 0.41% |
| Utensil | Eating | 89 | 0.34% | 16 | 0.14% | 12 | 0.20% | 29 | 0.59% | 32 | 0.93% |
| Food | Eating | 185 | 0.71% | 23 | 0.20% | 79 | 1.30% | 41 | 0.84% | 42 | 1.22% |
| Cellophane wrap | Eating | 128 | 0.49% | 42 | 0.36% | 28 | 0.46% | 8 | 0.16% | 50 | 1.45% |
| Train and bus ticket | Transport | 155 | 0.59% | 6 | 0.05% | 117 | 1.93% | 11 | 0.22% | 21 | 0.61% |
| Cash point or ATM receipt | Shopping and Entertainment | 272 | 1.04% | 93 | 0.79% | 39 | 0.64% | 120 | 2.45% | 20 | 0.58% |
| Till receipt | Shopping and Entertainment | 468 | 1.79% | 276 | 2.34% | 74 | 1.22% | 69 | 1.41% | 49 | 1.42% |
| Flyer and leaflet | Shopping and Entertainment | 374 | 1.43% | 55 | 0.47% | 212 | 3.49% | 45 | 0.92% | 62 | 1.80% |
| Cardboard box | Shopping and Entertainment | 52 | 0.20% | 19 | 0.16% | 22 | 0.36% | 3 | 0.06% | 8 | 0.23% |
| Newspaper and magazine | Shopping and Entertainment | 50 | 0.19% | 4 | 0.03% | 25 | 0.41% | 3 | 0.06% | 18 | 0.52% |
| Plastic bag | Shopping and Entertainment | 39 | 0.15% | 5 | 0.04% | 27 | 0.44% | 2 | 0.04% | 5 | 0.14% |
| Textile | Other | 18 | 0.07% | 4 | 0.03% | 3 | 0.05% | 4 | 0.08% | 7 | 0.20% |
| General litter and other | Other | 702 | 2.68% | 363 | 3.08% | 247 | 4.07% | 44 | 0.90% | 48 | 1.39% |

| | | | | | | | | | | |
|---|---|---|---|---|---|---|---|---|---|---|
| Unsure | Other | 86 | 0.33% | 6 | 0.05% | 41 | 0.67% | 11 | 0.22% | 28 | 0.81% |
| Bagged litter | Other | 72 | 0.27% | 35 | 0.30% | 32 | 0.53% | 5 | 0.10% | 0 | 0.00% |
| Total | | 26,209 | | 11,780 | 44.95% | 6075 | 23.18% | 4904 | 18.71% | 3450 | 13.16% |

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
