# Peer review of "Regional Variations in Urban Trash: Connections between Litter Communities and Place"

_sustainability, doi:10.3390/su16177741_

Round 1

Reviewer 1 Report

Comments and Suggestions for Authors

Abstract

·       Problem statement of the study not stated.

·       What is/are the problem with litter and littering in urban areas?

·       Objective of the study not stated.

Introduction

·       Problem statement which refer to the title was not given.

·       What is/are the problem(s) with litter and littering?

·       What is/ are the reference point of the litter and littering problems? Society? Behaviour? Technology? Urban design? System?

·       What is the purpose to determine litter profile?

·       Objective of the study was not given.

Methods

·       Since the problem statement and objective of the study were not clearly stated, hence the methods listed and described were found without a meaning.

·       Litter survey typology used were not described.

·       Why the litter typology lists were chosen?

·       Method for canvassing litter was not clearly stated for the four study sites.

·       What is the distance of litter canvassing conducted?

·       When was the litter canvassing conducted?

·       Does time and space influence litter survey?

·       What is/ are the methods use to collect data for community ecology modelling?

·       The method to measure the influence session on community parameter described was conducted to answer what?

·       There is no explanation on findings from each methods listed will be answering which problems.

·       What are the parameters used for:

o   Influence of session

o   Community analysis

Results.

·       There is a need to provide the litter and littering definition with reference to the aim of the study.

·       Results given in Table 5 is confusing. Cigarette is not an activity; it is a product. Smoking is an activity. Also same for Gum and Transport, these two also not an activity.

·       Author needs to define what is the parameter for activity and typology, which refer to object of littering.

·        

Reviewer 2 Report

Comments and Suggestions for Authors

Comment for the authors

Manuscript “Regional variations in urban trash: connections between litter communities and place”

In this study give an important overview about litter typology and effective parameters on it.

There are important findings about littering behavior and pattern.

The effect of location on litter is documented well.

Questions:

1-     How did authors select/determine the Community parameters? This part lacks clarity.

2-     Can authors elaborate this sentence a bit more?  “By framing litter as a product of place and not one of human behaviour, this study argues that sustainable waste management systems can be established on a local level.

Reviewer 3 Report

Comments and Suggestions for Authors

Comments on the Quality of English Language
